

# Determining the capacity for effort and recovery of the elite soccer players specialized in different playing positions

Liliana Mihailescu[1], Paul Bogdan Chiriac[2], Liviu Emanuel Mihailescu[3], Veaceslav Manolachi[4,5] and Vladimir Potop[1,3,4]

[1] Doctoral School of Sports Science and Physical Education, University of Pitesti, Pitesti, Romania
[2] Sports High School, Targu Jiu, Romania
[3] Department of Physical Education and Sport, University of Pitesti, Pitesti, Romania
[4] State University of Physical Education and Sport, Chisinau, Republic of Moldova
[5] "Dunarea de Jos" University of Galati, Galati, Romania

## ABSTRACT

**Background:** The capacity for effort and recovery in performance sports can be increased by means of modern methodological strategies. This capacity to recover after intense matches and training helps to determine the performance in soccer. Using the Yumeiho technique will accelerate the exercise capacity recovery of the soccer players specialized in central zone positions.

**Methods:** The research was conducted with the C.S. Internaţional Băleşti team, formed of 16 players, aged 19–37 years, divided into two groups: experimental group A, $n = 8$ with central area players and control group B, $n = 8$ with players from side zones. Measurement and assessment tests: determining lactic acid level (LAC) in blood; 60 m sprinting speed motor test (ST); anaerobic lactic exercise; Gacon Test for evaluating the aerobic power; VO2max test; Dorgo Test for determining the individual recovery capacity; Sleep Quality and Efficiency Index (PSQI) and sleep duration. Means used in the recovery strategy: easy running, static stretching, cryotherapy; Yumeiho therapy was applied only to group A to accelerate the recovery.

**Results:** The anaerobic lactic capacity was evaluated by 60 m sprint test at the beginning and the end of the research. The results highlight the increase of the sprinting speed by 0.08 s in group A ($p < 0.001$). The aerobic capacity evaluated by means of Gacon Test in both groups shows the value 23.7%, namely a well-prepared level ($p < 0.001$). VO2max value in the soccer players of group A shows 87.5% good aerobic power, while the players of group B have 50% good aerobic power ($p < 0.001$). The concentration of LAC after exercise has a higher value in group A ($p > 0.05$). The level of recovery after exercise is improved in both groups, with larger differences in group A (very good level, $p < 0.001$). The PSQI grew by 20.37% in group A and by 11% in group B. The sleep duration increased in both groups ($p < 0.001$). The results of the correlation analysis in the soccer players of group A highlight strong connections of 14.8% while in group B the value is 12.5%.

**Conclusion:** The assessment of effort capacity in soccer players specialized in different playing positions at the beginning and the end of the research highlights the increase of the anaerobic lactic effort and aerobic effort. This assessment also shows the improvement of the individual recovery capacity. A higher PSQI and the increase

Corresponding author
Liviu Emanuel Mihailescu, liviumihailescu2006@yahoo.com

of sleep duration in both groups, with greater differences of group A were found, which determined their level of capacity for effort and recovery. The use of the Yumeiho technique in the program of post-exercise recovery accelerates the aerobic and anaerobic lactic effort capacity of the soccer players specialized in the central zone positions. It positively influences exercise capacity recovery in general.

# INTRODUCTION

The increase of the performance capacity in elite sports is possible only through the use of modern specific and general methods/techniques and measures for recovery.
The relationship between training load, fatigue and the recovery-stress balance has received a considerable attention in the literature specialized in sports performance. The prevention of the non-functional overload and overtraining, the diminution of injury and illness risks were specifically addressed. Post-exercise recovery is related to the metabolic secondary products removal, the replenishment of energy stores and the initiation of tissue repairs (*Reilly & Ekblom, 2005*; *Barnett, 2006*; *Kozina, 2015*; *Thorpe et al., 2015*; *Bertollo et al., 2017*). Other authors believe that the recovery of exercise capacity is an integral part and essential component of the training process (*De Souza, Ribas & Lopes, 2017*). The variations and evolution of body composition, the neuromuscular and endurance-related parameters, also the game-related physical parameters in professional soccer players were approached. The aim was to find out how various training stimuli and situation variables affect the physiological and performance parameters of the players (*Silva, 2022*).

Recovery is considered as a multifaceted restorative process in relation to time. In case an individual's recovery status is disturbed by external or internal factors, fatigue as a condition of augmented tiredness due to physical and mental effort develops. Recovery and fatigue can be seen as a continuum and are jointly influenced by physiological and psychological determinants. Regeneration in sport and exercise refers to the physiological aspect of recovery and ideally follows physical fatigue induced by training or competition (*Kellmann et al., 2018*). The periods of repeated soccer-specific intense activities of aerobic and/or anaerobic nature entail serious strains on the physiological systems. These activities can cause declines and deteriorations in the performance, biological functions and perceptual responses of the players (*Bangsbo, Mohr & Krustrup, 2006*; *Silva et al., 2014*; *Reinke et al., 2009*; *Silva, 2022*). In the soccer game, there is a mixed-type effort (aerobic-anaerobic) in which three processes of energy release are involved: aerobic, non-lactic anaerobic and lactic anaerobic (*Bompa, 2013*; *Zubitashvili, Qobelashvili & Chkhikvishvili, 2013*; *Bekris, Mylonis & Gioldasis, 2016*). The active recovery is carried out according to a typical pattern with low volume and intensity, taking into account the current capacity and training load of each athlete. This recovery can be used immediately after high intensity training sessions, but also in competitions, especially when the anaerobic glycolysis process

is substantially involved (*Mihăilescu, 2011*). In the pre-season soccer training, morning and afternoon training sessions are often scheduled daily. The high frequency of training sessions could considerably place a strain on the biological systems. Therefore, it is necessary to use appropriate recovery strategies to improve the ability of the players to regain a suitable working condition for the following training units. The ability to recover after intense training, competitions and matches is considered a highly important factor in elite soccer. At the present moment there is no consensus regarding the effect of the post-exercise recovery interventions on the subsequent training sessions (*Tessitore et al., 2007*; *Rey et al., 2012*, *2018*; *Nédélec et al., 2013*, *2014*; *Pooley et al., 2017*).

The successful performance of the athletes depends on the training process quality and the timely medical support throughout training. Although research is limited on the effectiveness of various post-soccer match recovery methods, recovery in sports has become an increasingly important subject for coaches/physical trainers. The development of non-invasive methods for diagnosing the body reserve capacity is needed in sports practice to improve performance. Cryotherapy is generally used after soccer matches to accelerate the natural time-course of recovery. Another solution is the use of whole-body cryotherapy (WBC), namely a short exposure (2–3 min at the most) to dry air at cryogenic temperatures (up to −190 °C). The WBC has been applied for muscle recovery after injury for decreasing the inflammation process. The effects of a single WBC session performed shortly after repeated sprint exercise in professional youth soccer players were also investigated (*Russell et al., 2017*; *Brownstein et al., 2019*; *Moradi & Monazzami, 2020*; *Selleri et al., 2022*; *Martusevich et al., 2022*).

Yumeiho is a therapeutic and prophylactic technique meant to restore the biomechanical alignment in the myo-arthro-kinetic system, to relax the muscles and increase the conjunctive tissues elasticity. For this purpose, elements from stretching, osteopathy and different types of massage are used together. Through the effects induced at psycho-somatic and energetic level, Yumeiho technique helps to restore the biological and energetic structures affected by illness in athletes and non-athletes. This recovery is possible by increasing the capacity of the body to restore health. Yumeiho technique leads to the improvement of the local, articular and muscular vascularization (*Bratu, 2014*; *Bratu & Cordun, 2014*; *Chiriac, Mihăilescu & Bărbăcioru, 2021*).

Analyzing the importance of determining the effort capacity for the recovery capacity improvement in elite soccer players, the research intends to demonstrate that the use of Yumeiho therapy accelerates the post-exercise recovery in the players specialized in the central area positions compared to the lateral area positions. Generally, the Yumeiho technique has a positive influence on effort capacity recovery.

The purpose of the research is to determine the capacity for effort and recovery of the elite soccer players in different playing positions by associating the Yumeiho technique with other means of recovery.
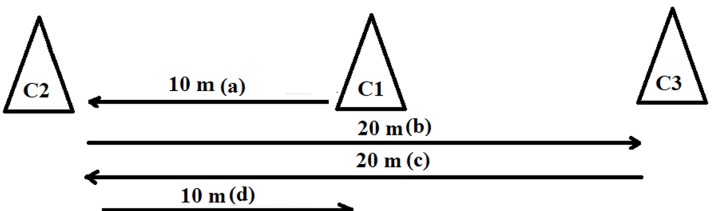

**Figure 1 Sprint starts from the middle marker cone (C1).** (A) 10 m sprinting up to C2 (C1–C2), (B) 20 m sprinting up to C3 (C2–C1–C3), (C) 20 m sprinting up to C2 (C3–C1–C2), (D) 10 m sprinting up to C1.                              

# MATERIALS AND METHODS

## Participants

This article is based on the case study of experimental longitudinal research conducted from July 2017 to June 2018, in the Sport Club (C.S.) Internațional Bălești team, formed of 16 players, aged 19–37 years. The aim was to prove that using Yumeiho therapy throughout the post-effort recovery program leads to the acceleration of effort capacity recovery in soccer players and positively influences the recovery of the effort capacity in general. The players were divided into two groups: group A (experimental group, $n = 8$): players from field central zone (defensive midfielders, center midfielders and attacking midfielder) and group B (control group, $n = 8$): players in the side zones (wingers and full-backs).

The experimental study was approved by the University of Pitesti Ethics Committee for the Doctoral School "Science of Sport and Physical Education" in accordance with the Ethical Principles in the Declaration of Helsinki (ecbr8-06-2017). The subjects gave written consent to the study in accordance with the recommendations of the Biomedical Research Ethics Committees (*e.g. WHO, 2000*). Given their age, the subjects presented also a written consent signed by their parents regarding their participation in the research.

## Instruments, apparatus, measurement and assessment tests

1) *Sprint motor test* (ST, s) on a total distance of 60 m (4 × 10; 20; 20; 10 m) (Fig. 1), considered anaerobic lactic exercise. A total of three marker cones were placed at a distance of 10 m from each other, each one in straight line, in the following order: 2, 1 and 3 (*Chiriac, Mihăilescu & Bărbăcioru, 2021*).

2) *Gacon test* consists of running a set distance within 45 s and resting 15 s, then repeating the sequence (25 phases in total). The initial distance to be run was 125 m and this distance increased by 6.25 m after each phase (for example: 125 m/131.25 m/137.5 m/ …); the running time remained unchanged (45 s). The total time of the test was 24 min 45 s and the total distance is 5,000 m. The main purpose of Gacon test is to assess the aerobic power of the subjects (VO2max, ml/min/kg) (*Albano, Serra & Vastola, 2019; Chiriac & Mihăilescu, 2019*).

*Gacon test*—interpretation:

– under 17 repetitions (phases)—very poorly prepared (under 2,975 m);

– between 18–22 repetitions (phases)—well prepared (3,206.25–4,193.75 m);
– between 23–25 repetitions (phases)—very well prepared (4,456.25–5,000 m).

Taking into consideration the specificity of the playing positions and the value of VO2max (ml/min/kg), the physical trainers involved in the research developed the following interpretation:

*Full-backs/wingers*:

– 56—very poor aerobic power;
– 57—poor aerobic power;
– 59—satisfactory aerobic power;
– from 60 to 62—good aerobic power

*Defensive midfielders*:

– from 56 to 57—good aerobic power

*Central midfielders*:

– from 57 to 59—poor aerobic power;
– from 59 to 60—satisfactory aerobic power;
– from 60 to 62—good aerobic power.

*Attacking midfielder*: from 59 to 60—good aerobic power.

3) The level of lactic acid (LAC) in blood was measured by means of LactatePro 2 (LT-1730; ARKRAY, Kyoto, Japan) device. The analysis was done by the electrochemical method, using an enzymatic reaction. The blood (a quantity of 0.3 µL only) was collected through the capillarity phenomenon. The analysis automatically started when the blood quantity of 0.3 µL enters the capillaries (the device does not have a start button) and the result was given in 15 s. LAC was evaluated immediately after exercise in each individual athlete, according to the test performed (*Crotty et al., 2021*).

4) *Dorgo test* for determining the individual recovery capacity of the players, calculating the specific recovery index after continuous exercise. The Dorgo test highlights the individual cardiac behavior during exercise by means of the heart rate values (HR). The individual values of heart rate (bpm) were monitored as follows: pulse before exercise (P1) and in the minutes 2, 4 and 6 post exercise (P1-4) (*Chiriac, Mihăilescu & Bărbăcioru, 2021*).

$$ID = (P1 + P2 + P3 + P4) - 300/10$$

Interpretation of values: −5 and −10 = very good index; −4 to 0 = good index; 0–5 mediocre index; 6–10 = poor index, over 10 = very poor index.

5) The quality, efficiency and duration of the sleep were determined by means of Samsung Gear S3 Frontier smartwatches (Released 2016, Tizen OS 5.5; 4 GB, 768 MB RAM storage). Sensors included GPS, barometer, HRM, ambient light sensor. The benefits of these sensors are that the LTE radio periodically tracks the fitness data. Sleep Quality

**Table 1 Content of recovery used throughout the experiment.**

| No. | Content of the recovery | Purpose | Amount | Methodical indications |
|---|---|---|---|---|
| 1. | Easy running —pulse 120–130 (bpm) | Maintaining the heart rate above the resting rate and increasing lactic acid metabolism | One round of 10 min continuous run | Maintaining the imposed pace |
| 2. | Static stretching | Improvement of blood circulation at muscle level, relaxing tense muscles and inducing a general state of well-being by stress diminution | Two rounds of 5 min with 30 s of muscle stretching and 30 s of pause. | Bringing the joint to an angle where the musculature is at its maximum stretching point; holding it for a few seconds in this position |
| 3. | Cryotherapy (Ice bath at a temperature of +10–15 °C) | Cooling the muscles involved in the effort, favoring circulation and reducing the discomfort created by muscle and joint pain | Three rounds of 30 s each —immersion in the ice pool/30 s pause | Staying in the pool until the sensation of numbness appears, but no more than 30 s. |
| 4. | Yumeiho therapy | Removing fatigue, relaxing the muscles and increasing tissue elasticity | | Correct execution of maneuvers |

and Efficiency Index was automatically calculated by the device depending on the settings made later for each athlete separately. According to the data obtained, Sleep Quality and Efficiency Index (PSQI) was scored as follows: Very well (100–90%), Well (89–80%), Average (79–70%), Poor (68–60%) and Very poor (>59%). The duration of the sleep was measured in hours: minutes, then transformed into minutes and seconds (*Robey et al., 2014*).

## Experimental design

The research was carried out in three stages:

*The first stage* (initial testing), corresponding to the preparatory stage, monitored the level of the indices assessed at the beginning of the research, namely: 60 m sprint (s); post-exercise LAC (mmol/L) after 60 m sprint test (the Romanian Football Federation—FRF—recommended six repetitions with maximum intensity intervals), Gacon test and VO2max, Dorgo index test (points), PSQI (%) and sleep duration (min).

*The second stage* applied the methods for the recovery strategy (Table 1):

The recovery strategy was implemented in 20 sessions (1 session/week) from July to November 2017, with the following objectives: to remove the fatigue, to relax the muscles, to cool down the muscles involved in effort, to reduce the discomfort created by the muscular and articular strain.

Other purposes: to increase the tissues elasticity, to maintain the heart rate above the resting rate, to increase the lactic acid metabolism, the improve the blood circulation in muscles, to induce a general state of well-being (by stress diminution).

*The third stage* (final testing) was meant to determine the assessed indices level at the end of the research: 60 m sprint (s); post-exercise LAC (mmol/L) after 60 m sprint test (the FRF recommended 10 repetitions with intervals at maximum intensity), Gacon test and VO2max, Dorgo index (points), PSQI (%) and sleep duration (min), LAC (mmol/L) after

using various recovery methods (easy running, static stretching and criotherapy) in both groups, plus Yumeiho techniques used in group A only.

Table 1 shows the main means and techniques of recovery, specifying the purpose, amount and methodical indications related to easy running (10 min) with pulse maintained at 120–130 bpm, static stretching (2 × 5 min × 30 s of muscle stretching and 30 s of pause), cryotherapy (ice bath at a temperature of +10 … −15 degrees Celsius)−3 × 30 s immersion in the ice pool × 30 s pause and Yumeiho therapy used for group A in 19 sessions with an average duration of 38–40 min.

## Recovery strategy used in research

The $3^{rd}$ League Football National Championship, 2017–2018 season, was held on a double round-robin basis, namely 14 stages home and 14 stages away. This was the competitive period. During the research, all the players who participated in the Championship return matches benefitted from a series of classic recovery means (massage, cold water pool, running, cryotherapy). The Yumeiho technique was used for the players of group A only. The recovery process was conducted using a differentiated strategy in the two groups throughout the research.

For the selection and application of the recovery procedures, it was considered that the effort in the sprint test was of anaerobic lactic type, a characteristic effort of the competitive period that was the object of the experimental research. The recovery aimed at reducing the LAC accumulated in the muscles. The maneuvers of Yumeiho therapy used in the post-exercise recovery are summarized in Table 2.

## Statistical analysis

Data analysis was made using Statistical Package for Social Sciences (SPSS-PC) version 20 software. Descriptive statistics methods were applied to calculate: the median and standard deviation (SD); Cohen's d effect size; t-Test, Unpaired Comparison for Means—between groups; t-Test, Paired Comparison for Means—between testing stages; $X^2$—Friedman Test with Replication between LAC means after the different recovery methods used; Rho—Spearman Rank Correlation between the recovery capacity indices before and after applying the recovery methods. The level of confidence was set to 95% and considered statistically significant ($p$ value ≤ 0.05).

## RESULTS

The experimental research started with an assessment of the players of C.S. Internaţional Băleşti soccer team, which determined their capacity for effort and recovery highlighted by the following indices: 60 m S.T.; Gacon test; post-exercise LAC; Dorgo index; sleep quality and efficiency index; sleep duration.

The results of the assessment of the capacity for effort and recovery in the soccer players specialized in different playing positions are listed in Table 3. The assessment was made at the beginning of the research and at the end of this one.

The analysis of the results of this assessment shows significant differences between tests at ST (60 m), namely 0.6 s in group A ($p < 0.001$), 0.68 s in group B ($p < 0.001$), effect size

**Table 2 Maneuvers of the Yumeiho technique used in group A.**

| No. | Type of intervention | Areas involved | Initial position | Description of the exercise | Amount | Pause duration |
|---|---|---|---|---|---|---|
| 1. | Osteopathy | Scapulo-humeral joints and upper thoracic vertebrae | On the knees, sitting on the heels, with the hands behind the head | The kinesiotherapist, staying behind the athlete, in kneeling position, with thighs abducted, grasps the pelvis of the subject, executes an anterior grip on the forearms and performs horizontal adduction of the arms, simultaneously with the extension and traction of the spine. At the point of maximum tension, the manipulative impulse is induced by raising the shoulders and extending the arms at the same time with the anterior projection of the abdomen. | One repetition at the maximum point | Passive for 10 s. |
| 2 | Stretching | The anterior muscles of the thighs and calves | Ventral decubitus position | The kinesiotherapist, on his knees, grasps the tops of the athlete's feet with both hands and passively performs the legs extension on the calves and the calves flexion on the thighs. At the point of maximum amplitude, he executes a prolonged final tension through the pressure exerted on the dorsal side of the legs. | One repetition at the maximum point | Passive for 10 s. |
| 3 | Massage | Left/right biceps femoris muscle tendon | Ventral decubitus position | The kinesiotherapist executes linear friction perpendicular to the tendon of biceps femoris (at the level of gluteal fold, situated beneath the ischium) through "cutting maneuvers" performed with both thumbs, placed perpendicularly on top of each other. | 10 points on the same line | Passive for 10 s. |
| 4. | Massage | Left/right thigh (back side) | Ventral decubitus position | The kinesiotherapist massages the region between the gluteal fold and the popliteal fossa, along three lines, a median one, a medial one and a lateral one. The maneuver consists of linear friction with kneading, in 8–10 points/line with both thumbs, placed perpendicularly on top of each other | - 8–10 points on the midline, two times; - 8–10 points on the medial line, two times ; - 8–10 points on the lateral line, two times | Passive for 10 s. after each line |
| 5. | Massage | Left/right calf (back side) | Ventral decubitus position | The kinesiotherapist performs massage from the popliteal fossa up to the Achilles tendon, along three lines, a median one, a medial one and a lateral one. The maneuver consists of linear friction with kneading, in 6–8 points/line with both thumbs, placed perpendicularly. | - 6–8 points on the midline, two times; - 6–8 points on the medial line, two times; - 6–8 points on the lateral line, two times | Passive for 10 s. after each line |
| 6. | Massage | Triceps surae muscles | Ventral decubitus position, knees extended, legs in inversion | The kinesiotherapist, on his knees, kneads simultaneously the back sides of the calves | In 5 points, two times | Passive for 10 s. |

| No. | Type of intervention | Areas involved | Initial position | Description of the exercise | Amount | Pause duration |
|-----|---------------------|----------------|------------------|----------------------------|--------|----------------|
| | Table 2 (continued) | | | | | |
| 7. | Stretching | Adductor muscles and coxofemoral joints (flexion—abduction—external rotation) | Dorsal decubitus position, with the thighs in flexion—abduction—external rotation, knees flexed, sole into sole | The kinesiotherapist presses continuously the medial surfaces of the knees, initially alternately, then simultaneously. | In 6–8 points, two times | Passive for 10 s. |
| 8. | Massage | Left/right thigh (front side) | Dorsal decubitus position, with the left thigh slightly abducted and externally rotated | The kinesiotherapist performs: a) massage of the inner side of the thigh, along three lines—median, medial and lateral one; b) massage of the tensor muscle of fascia lata, with both thumbs placed perpendicularly on top of each other; c) passive mobilization of the patella | - In 10 points on the median line, two times - In 10 points on the medial line, one time - In 10 points on the lateral line, one time | Passive for 10 s. after each line |
| 9. | Massage | Posterior musculature of left/right thigh and calf | Dorsal decubitus position | The kinesiotherapist, on his knees, sitting on the heels, perpendicular to the patient's long axis, rolls the lower limb of the patient on his thighs | 6–8 reps | Passive for 10 s. |
| 10. | Massage | Left/right anterior tibialis muscle | Dorsal decubitus position | The kinesiotherapist massages the tibialis anterior muscle with both thumbs, placed perpendicularly on top of each other | In 5–6 points, two or three times | Passive for 10 s. |

(ES) medium between groups and very large between tests, with higher values in group A and insignificant differences between groups ($p > 0.05$). The value of Gacon Test in both groups is 23.7% up to the total distance (5,000 m), ES medium (initial) and very large in group A (final) and large in group B, indicating a well-prepared level ($p < 0.001$) and insignificant differences between groups ($p > 0.05$). LAC concentration after exercise has a higher value in group A, insignificant differences between tests of 0.76 mmol/L in group A and of 1.37 mmol/L in group B ($p > 0.05$) with ES large between groups and small between tests in both groups. VO2max value in the soccer players from the central zone (group A) is 87.5% good aerobic power, ES small between groups, large in group A and medium—group B. Soccer players in the lateral zone (group B) have 50% good aerobic power; there are significant differences between tests in both groups ($p < 0.001$). The level of recovery after continuous exercise improved in both groups with greater differences in group A, namely 12.17 points (very good level, $p < 0.001$) and 6.52 points in group B (average level, $p > 0.05$) with very large ES between groups and between tests, with larger values in group A. The index of sleep quality and efficiency increased by 20.37% in group A and 11% in group B cu very large ES between groups and between tests, with larger values in group A.

An increase in the sleep duration was noticed as follows: group A—by 68.12 min and group B—by 27.5 min, ES small (initial) and very large (final) between groups and between tests, with higher values in group A. Significant differences between tests and between

**Table 3 Results of the capacity for effort and recovery of the soccer players specialized in different playing positions at the beginning and the end of the research.**

| Indices assessed | Stages, statist. indic. | Median ± SD | | Cohen's d | t | p-value |
|---|---|---|---|---|---|---|
| | | Group A | Group B | | | |
| 60 m S.T. (s) | Initial | 12.06 ± 0.27 | 12.23 ± 0.39 | −0.51 | −0.99 | 0.339 |
| | Final | 11.35 ± 0.31 | 11.44 ± 0.27 | −0.31 | −0.61 | 0.553 |
| | Cohen's d | 2.06 | 2.03 | – | – | – |
| | t; p-value | 6.31; 0.000*** | 5.57; 0.008*** | – | – | – |
| Gacon test (m) | Initial | 3,446.9 ± 284.4 | 3,687.5 ± 259.3 | −0.21 | −0.43 | 0.673 |
| | Final | 3,812.6 ± 298.8 | 3,937.5 ± 266.03 | 0.003 | 0.006 | 0.996 |
| | Cohen's d | −1.05 | −0.94 | – | – | – |
| | t; p-value | −7.72; 0.000*** | −104.5; 0.000*** | – | – | – |
| Post-exercise LAC (mmol/L) | Initial | 20.8 ± 2.78 | 19.0 ± 3.81 | 0.69 | 1.40 | 0.182 |
| | Final | 21.1 ± 2.21 | 20.3 ± 3.41 | 0.59 | 1.19 | 0.253 |
| | Cohen's d | −0.30 | −0.38 | – | – | – |
| | t; p-value | −1.16; 0.283 | −1.71; 0.130 | – | – | – |
| VO2 max (ml/min/kg) | Initial | 58.0 ± 1.69 | 59.0 ± 1.55 | −0.08 | −0.15 | 0.879 |
| | Final | 59.5 ± 1.93 | 60.0 ± 1.68 | 0.07 | 0.14 | 0.892 |
| | Cohen's d | −0.83 | −0.77 | – | – | – |
| | t; p-value | −7.94; 0.000*** | −7.64; 0.000*** | – | – | – |
| Dorgo index (points) | Initial | 6.3 ± 1.16 | 8.7 ± 1.87 | −1.16 | 2.315 | 0.036* |
| | Final | −5.25 ± 0.19 | 2.18 ± 0.46 | −21.17 | −42.28 | 0.000*** |
| | Cohen's d | 14.64 | 4.79 | – | – | – |
| | t; p-value | 27.25; 0.000*** | 1.14; 0.292 | – | – | – |
| PSQI (%) | Initial | 72.5 ± 2.26 | 71.5 ± 2.45 | 0.37 | 0.83 | 0.44 |
| | Final | 92.5 ± 1.28 | 82.5 ± 1.60 | 7.07 | 14.12 | 0.000*** |
| | Cohen's d | −11.09 | −5.32 | – | – | – |
| | t; p-value | −22.51; 0.000*** | −9.32; 0.000*** | – | – | – |
| Sleep duration (min) | Initial | 422.5 ± 17.92 | 416.25 ± 26.96 | 0.08 | 0.26 | 0.81 |
| | Final | 485.0 ± 9.16 | 447.5 ± 11.87 | 4.01 | 8.01 | 0.000*** |
| | Cohen's d | −4.79 | −1.32 | – | – | – |
| | t; p-value | −14.74; 0.000*** | −4.34; 0.003** | – | – | – |

**Notes:**
\* $p < 0.05$.
\*\* $p < 0.01$.
\*\*\* $p < 0.001$.
S.T., sprinting test; LAC, lactic acid; PSQI, sleep quality and efficiency index; SD, standard deviation; Cohen's d effect size (Small 0.2 Medium 0.5 Large 0.8 Very large 1.3); t-Test. Unpaired Comparison for Means—between groups; t-Test. Paired Comparison for Means—between testing stages.

groups were observed at the final test ($p < 0.001$) and insignificant differences between groups were seen at the beginning of the research ($p > 0.05$).

The results regarding the level of lactic acid accumulated in blood after various methods of recovery used in the soccer players specialized in different playing positions at the end of the research are presented in Table 4.

This analysis highlights a higher value of lactate (LAC)—by 1.27 mmol/L—in group A after performing an easy run of 10 min, maintaining the pulse at 120–130 (bpm). LAC

**Table 4 The level of lactic acid accumulated in blood after different recovery methods used in the soccer players specialized in various playing positions at the end of the research.**

| Indices assessed | Median ± SD | | Cohen's d | t | p-value |
|---|---|---|---|---|---|
| | Group A | Group B | | | |
| LAC after running (mmol/L) | 16.5 ± 1.64 | 15.2 ± 6.56 | 0.266 | −1.174 | 0.259 |
| LAC after stretching (mmol/L) | 14.8 ± 1.48 | 13.65 ± 5.27 | 0.295 | 1.177 | 0.258 |
| LAC after cryotherapy (mmol/L) | 12.5 ± 1.24 | 11.55 ± 3.85 | 0.336 | 1.171 | 0.261 |
| LAC after yumeiho (mmol/L) | 9.3 ± 0.91 | – | – | – | – |
| $X^2$ | 24 | 16 | – | – | – |
| p-value | 0.0005*** | 0.0003*** | – | – | – |

Notes:

*** $p < 0.001$.

SD, standard deviation; Cohen's d effect size (Small 0.2 Medium 0.5 Large 0.8 Very large 1.3); t-Test. Unpaired Comparison for Means; $X^2$, Friedman Test with Replication.

value increased by 1.14 mmol/L after stretching and by 0.96 mmol/L ($p > 0.05$) after cryotherapy. Effect size is small between groups. After using Yumeiho therapy in group A besides the other classic methods of recovery applied to both groups, a decrease by 2.15 mmol/L in LAC concentration is noticed in group A compared with group B.

The connections between the recovery capacity indices, before and after applying the classic recovery means (group A and B) and adding Yumeiho technique to group A, were highlighted by Spearman's nonparametric correlation analysis. The results regarding the correlations are shown in Figs. 2 and 3.

The connections between the eight indices assessed at the research beginning and the 11 indices assessed at the end, after applying the recovery methods and associating Yumeiho therapy, were analyzed. A total of 88 corelations were made (36.4% negative and 63.6% positive ones), out of which 14.8% strong connections as follows: between Gacon Test (initial-final), VO2max (initial-final) and between their indices at $p < 0.001$; between 60 m S.T. (initial-final); between post-exercise LAC (initial-final). There are strong connections at $p < 0.001$ between tests and the indices assessed (4.5%). Also, strong connections were found out at $p < 0.05$ between tests (2.3%) and the indices assessed (8.0%).

There were examined the connections between the eight indices assessed at the beginning of the research and the 10 ones assessed at the end, following the application of the classic methods of recovery. A number of 80 correlations were made (25% negative and 75% positive), out of which 12.5% strong connections. It was noticed that 6.25% are at $p < 0.001$ between tests and indices in Gacon Test and VO2max and between tests at Dorgo index. As for $p < 0.05$ there are 6.25% between the indices of 60 m S.T. and VO2 max, between post-exercise LAC after 60 m S.T. and the means of recovery (after running, after stretching and after cryotherapy).

After carrying out the correlation analysis in the soccer players specialized in different playing zones, strong connections were observed between post-exercise LAC after 60 m S.

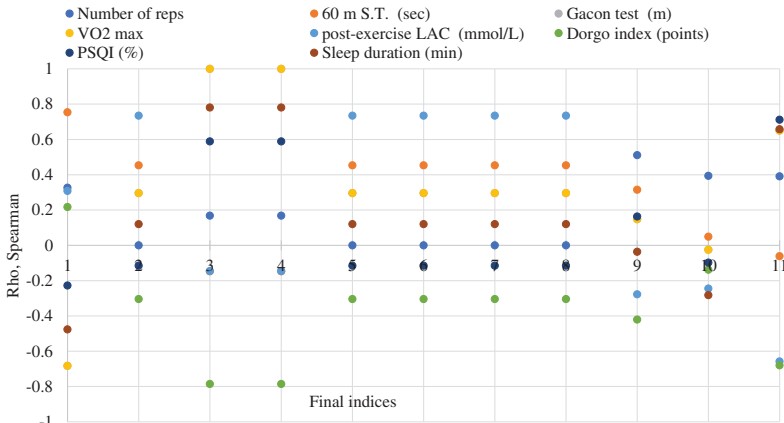

**Figure 2 Correlation analysis between the recovery capacity indices before and after the application of the recovery means and the use of the Yumeiho technique in the soccer players specialized in the central playing zone (group A).** Initial indices (1–8): legend; final indices (1–11): 1–60 m ST (s). 2-post-exercise LAC (mmol/L). 3-Gacon Test (m); 4-VO2 max (ml/min/kg); 5-LAC after Running (mmol/L). 6-LAC after Stretching (mmol/L). 7-LAC after Cryotherapy (mmol/L). 8-LAC after Yumeiho (mmol/L). 9-Dorgo Index (points). 10-Sleep quality (%). 11-Sleep duration (min).

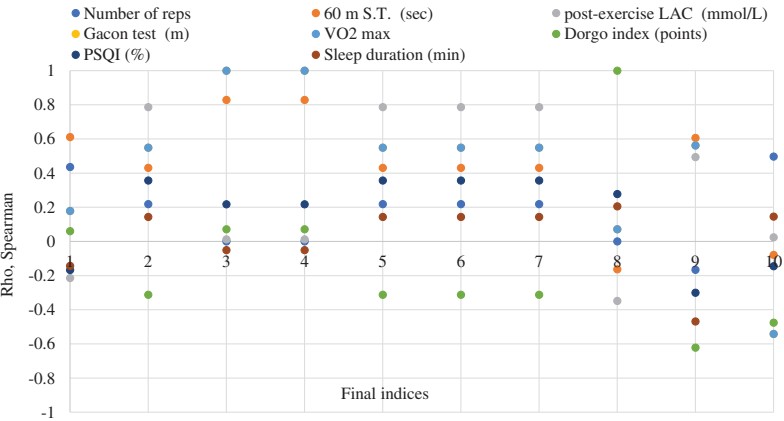

**Figure 3 Correlation analysis between the indices of the capacity for recovery before and after the application of the recovery means in the soccer players specialized in the lateral zones of playing (group B).** Initial indices (1–8): legend; final indices (1–10): 1–60 m ST (s). 2-post-exercise LAC (mmol/L). 3-Gacon Test(m); 4-VO2 max (ml/min/kg); 5-LAC after Running (mmol/L). 6-LAC after Stretching (mmol/L). 7-LAC after Cryotherapy (mmol/L). 8-Dorgo index (points). 9-Sleep quality (%). 10-Sleep duration (min).

T., considered anaerobic lactic exercise, and Gacon Test—aerobic exercise, between the means of classic recovery used in both groups and Yumeiho therapy used in group A.

## DISCUSSION

The research was meant to determine the capacity for effort and recovery of the soccer players specialized in different playing zones. For this purpose, assessment specific motor tests and strategic methods of recovery were used. Analyzing the results of the anaerobic lactic exercise capacity (assessed by 60 m S.T. at the beginning and the end of the research),

the increase of the sprinting speed by 0.08 s in group A can be noticed. The results of the aerobic capacity assessed by Gacon test in both groups show a well prepared level of 23.7%. In terms of running speed, some studies reveal the effect that a combined endurance and running speed workout performed in the same session has on the strength, speed and vertical jumping of the soccer players (*Kotzamanidis et al., 2005*). Other studies examined the use of a customized high-speed threshold according to the speed at the second ventilatory threshold (VT2speed) for assessment of the high-intensity distance running along matches (*Abt & Lovell, 2009*). It was also investigated how a 10-week training program with frequent 40 m sprint workouts but without strength exercises can influence the maximum sprinting speed of the elite youth soccer players (*Tønnessen et al., 2011*). There are specialists who analyzed the impact of the maximal sprinting speed (MSS) on the peak speed reached during the soccer matches (*Mendez-Villanueva et al., 2011*). Other specialists analyzed the influence had by 4 weeks of training based on speed endurance maintenance (SEM) and speed endurance production (SEP) on the 5-m multiple shuttle run test (5-m MST) in the case of young soccer players (*Vitale et al., 2018*). It was also compared the training and match load of the professional soccer players depending on their playing position, analyzing the relationship between the metabolic and running speed metrics (*Guerrero-Calderón et al., 2022*).

VO2max value highlights a good aerobic power in both groups, namely 87.5% in the soccer players of the central zone (group A) and 50% in the soccer players of the lateral zone (group B). These values have a very big impact on the speed and the level of regeneration after training, mainly in the case of maximum intensity efforts. Regarding VO2max, the relationship between laboratory-measured VO2max and the total distance covered in a soccer match at different running intensities was assessed. VO2max value helped to quantify the intensity running in different playing positions and to determine the running differences between the game halves (*Metaxas, 2021*).

The analysis of the post-exercise recovery capacity at 60 m S.T. after 10 repetitions, at the end of the research, shows that the lactate concentration has a higher value in group A, namely 0.76 mmol/L; group B has a value of 1.37 mmol/L. It was also examined whether only one soccer specific fitness test—according to the selected performance indicators—could distinguish between elite soccer players and recreationally active ones (*Edwards, Macfadyen & Clark, 2003*). The training load applied throughout a one-year training period entailed changes in both aerobic and anaerobic physical capacity as well as in sport-specific skills of soccer players (*Jastrzebski et al., 2011*). The effect of the fatigue of the lower limbs muscles on the proprioceptive sense in soccer players was studied. The lactic acid concentration of the players at rest and their proprioceptive sense measurements were determined by standardized materials and methods (*Göktepe et al., 2019*). The reliability of the portable Lactate Pro 2 analyzer (LP2) and its validity compared with a laboratory-based analyzer, YSI 1500 Sport (YSI), were evaluated (*Crotty et al., 2021*). Other subject approached by the specialists was the effect of cold-water immersion during halftime on the physical recovery of the soccer players. The conclusion was that this immersion in cold water during halftime reduced muscle fatigue and decreased lactic acid

level. This procedure can be used as an alternative recovery method for soccer players (*Panyakham & Pariwat, 2022*).

The individual capacity of the players to recover after exercise was assessed by means of Dorgo test. The results highlight greater differences in both groups, as follows: 12.17 points in group A (very good level) and 6.52 points in group B (average level) (*Chiriac, Mihăilescu & Bărbăcioru, 2021*).

The assessment of the quality, efficiency and duration of the sleep reveals the index increasing by 20.37% in group A and by 11% in group B. The duration of the sleep increased by 68.12 min in group A and by 27.5 min in group B. In this sense, studies were carried out relating to the role of sleep in the recovery. Future research is needed to estimate the quantitative and qualitative importance of sleep, to identify the influencing factors and to find efficient and individualized solutions. The relationship between the frequency of the playing actions performed during a soccer match and the recovery structure after the match was also examined (*Clemente et al., 2021*; *Nédélec et al., 2013*, *2014*, *2015a*). The effect of early evening high-intensity training on the sleep of elite male youth soccer players was studied as well. The specialists found out that this type of training had no impact on the quality and quantity of the subsequent sleep (*Robey et al., 2014*). In elite soccer, players are often subject to various conditions and situations that can possibly lead to sleep deprivation. Therefore, it may be necessary to take into account efficient and individualized strategies of sleep hygiene (*Nédélec et al., 2015b*). A similar topic was approached by the specialists who assessed the sleep characteristics of the professional soccer players in Qatar Stars League. A poor sleep quality was reported in their case (68.5% only), which should be a real concern for the persons involved. It is very important to take into consideration the role of the sleep related to sports performance, injuries, illness and recovery (*Khalladi et al., 2019*). The sleep hygiene (SH) education has a considerable influence on the sleep quality of the soccer players after late-evening workouts. Soccer players may benefit from SH strategies to decrease the time necessary for falling asleep after such late training sessions (*Vitale et al., 2019*). Some authors studied the habitual sleep and nocturnal cardiac autonomic activity (CAA) and their relationship with the training and match load in male youth soccer players throughout an international tournament (*Figueiredo et al., 2021*).

The results regarding the lactic acid (LAC) accumulated in the blood after different classic recovery methods used in the soccer players at the end of the research show a higher value of LAC in group A. After using Yumeiho therapy in group A, a decrease in LAC concentration was observed, which entailed the diminution of the recovery time. Numerous studies investigate the efficacy of a single exposure to 14 min of contrast water therapy (CWT) or cold-water immersion (CWI) during the recovery after matches of the elite professional soccer players. An elite professional football match is followed by physical and psychometric deficits that last 48 h. CWI had better results in restoring physical performance and psychometric measures than the CWT (*Elias et al., 2013*). A comparison of three post-match recovery methods was made to find out their effects on the physical performance, physiological measures and perceptions of the players regarding the recovery after a soccer match of 90 min. The positive effects on perceived recovery after

the combined methods indicate that this approach may be efficient in the case of the youth players after intense soccer matches (*Kinugasa & Kilding, 2009*). Several strategies of recovery are currently used in the professional soccer teams for reducing the fatigue level and accelerating the time for full recovery. In this respect, the fatigue resulted from soccer match playing, the recovery kinetics of physical performance and the cognitive, subjective and biological markers were analyzed (*Pooley et al., 2017*). Two groups, each one formed of 10 subjects, were submitted to cryotherapy (10 min cold water immersion, 10 °C, $n = 10$) and thermoneutral environment (10 min thermoneutral water immersion, 35 °C, $n = 10$). The purpose was to assess the effects of a single session of cold or thermoneutral water immersion after a single match on the muscular dysfunction and damage in soccer players. The results indicate that cold water immersion right after the match decreases muscle damage and discomfort and contributes to a quicker recovery of the neuromuscular function (*Ascensão et al., 2011*). Wearing lower-body clothing provided with cooled phase change material can be a solution to improve the recovery of soccer players following a match (*Clifford et al., 2018*). Other specialists too examined the effect of the garments with phase change material on the recovery of the neuromuscular function after a competitive soccer match (*Brownstein et al., 2019*).

The active recovery methods and their effects on performance and muscle damage indices in young male soccer players were also studied. The foam rolling can be more successfully used to speed the recovery in comparison with water immersion (*Moradi & Monazzami, 2020*). The specialists analyzed how different levels of whole-body cryotherapy (WBC) exposure influence the subjective and objective measures of recovery after matches in the elite soccer players (*Malone et al., 2021*). It was determined the impact of the cryotherapy on the immunological, hormonal and metabolic responses in non-professional soccer players (NPSPs) (*Selleri et al., 2022*). Research was conducted on the crystalloscopic characteristics of oral fluid during various periods of the training and competitive cycle. It was proved that the biocrystallization test results are reliable and provide the necessary information about the regulatory mechanisms which influence the performance of athletes (*Bocharin et al., 2022*).

The results of the correlation analysis between the indices assessed at the beginning and at the end of the research on soccer players highlight 14.8% strong connections in group A (central zone) and 12.5% in group B (lateral zone).

The addition of Yumeiho technique to the classic means of recovery specific to soccer players specialized in the central zone ensures a higher level of recovery. This fact is revealed by the significant differences between the research groups, recorded at all parameters investigated.

Therefore, it is confirmed that applying Yumeiho technique in addition to the classic means of recovery specific to soccer players enables an efficient recovery of the anaerobic lactic capacity and of the aerobic one. In support of this statement, the differences between the average values recorded in group B should be taken into consideration. These differences show that the recovery time decreases.

## CONCLUSIONS

Exercise capacity assessment carried out at the beginning and the end of the research on the soccer players specialized in different playing zones highlights an increased anaerobic lactic capacity and aerobic capacity. This assessment also shows an improved individual post-exercise recovery capacity. A higher sleep efficiency and quality index and the increase of sleep duration in both groups, with bigger differences in group A, were discovered, which determined the level of the capacity for effort and recovery of the players.

Analyzing the results of the post-exercise recovery capacity after different classic methods of recovery used at the end of the research points out a higher concentration of lactic acid in group A. After adding Yumeiho therapy to group A besides the other recovery classic methods used in both groups, a decrease of the lactic acid concentration was noticed, which helped to reduce the post-exercise recovery time.

The correlational analysis performed for these soccer players shows strong connections between LAC level after sprinting, considered anaerobic lactic exercise, and Gacon Test—aerobic exercise, also between the classic recovery methods in both groups and Yumeiho therapy in group A.

Using the Yumeiho technique in the post-exercise recovery program enables a faster recovery of the anaerobic lactic capacity and aerobic capacity of the soccer players specialized in central zone playing positions. It has also a positive influence on the recovery of the exercise capacity in general.

## LIMITATIONS OF THE STUDY

Although there were numerous positive aspects of the training, however the study has some limitations. One of them is the fact that the experiment was conducted with a single soccer team, constituted as a case study.

Other limitation is that Yumeiho therapy—specific to kinesiology—was not used in other research and studies related to performance soccer. Therefore, it was not possible to make comparisons with the results of other research aiming at the recovery of athletes.

Also, the capacity of the subjects to objectively assess the impact of Yumeiho therapy in the context of the training sessions and official matches is a limitation.

## ACKNOWLEDGEMENTS

This study is part of the research plan topics of the Sports Science and Physical Education doctoral school in the University of Pitești. We express our gratitude to the C.S. Internaţional Băleşti director, to the technical training staff, to physical trainers Dumitru Vasilică (FCU Craiova) and Coc Sergiu Raul (CFR Cluj 2) for the interpretation of the collected data. Last but not least we thank the soccer players who participated in this study.

### Funding

The authors received no funding for this work.

## Grant Disclosures

The following grant information was disclosed by the authors:
The authors received no funding for this work.

## Competing Interests

The authors declare that they have no competing interests.

## Author Contributions

- Liliana Mihailescu conceived and designed the experiments, performed the experiments, analyzed the data, prepared figures and/or tables, authored or reviewed drafts of the article, and approved the final draft.
- Paul Bogdan Chiriac conceived and designed the experiments, performed the experiments, analyzed the data, prepared figures and/or tables, authored or reviewed drafts of the article, and approved the final draft.
- Liviu Emanuel Mihailescu performed the experiments, analyzed the data, prepared figures and/or tables, authored or reviewed drafts of the article, and approved the final draft.
- Veaceslav Manolachi conceived and designed the experiments, performed the experiments, authored or reviewed drafts of the article, and approved the final draft.
- Vladimir Potop conceived and designed the experiments, performed the experiments, analyzed the data, prepared figures and/or tables, authored or reviewed drafts of the article, and approved the final draft.

## Human Ethics

The following information was supplied relating to ethical approvals (*i.e.*, approving body and any reference numbers):

University of Pitesti Ethics Committee for the Doctoral School "Science of Sport" (ecbr8-06-2017).

## Data Availability

The raw measurements are available in the Supplemental File.

## Supplemental Information

Supplemental information for this article can be found online at http://dx.doi.org/10.7717/peerj.15477#supplemental-information.

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
