# Peer review of "Determining the capacity for effort and recovery of the elite soccer players specialized in different playing positions"

_PeerJ, doi:10.7717/peerj.15477_

## Round 0.1 · original submission · Major Revisions

Please, carefully address all Reviewer 2, 3 and 4's (in particular Reviewer 2's) issues.

·

Basic reporting

The article topic is relevant. It is actually for all dport games.

Experimental design

Experemental design is right.

Validity of the findings

Validity

Reviewer 2 ·

Basic reporting

As clearly stated in the title, the authors aimed to determine the capacity for effort and recovery of the elite soccer players specialized in different playing positions. However, I was not able to find the same clarity throughout the manuscript (abstract included). For example, reading the introduction I am not sure the authors are trying to determine or improve the capacity they mention in the title. This should be mandatorily clarified. Moreover, the abstract lacks an explanation, even simple, of what Yumeiho technique is and the introduction does not comprehensively describe it. Unfortunately, the topic of recovery is very vague too and makes the manuscript, at least in my opinion, not adequately framed (as an example I report here one of the sentences in this regard: “Post-exercise recovery occurs after the ending of exercise”). I suggest the authors read the literature on recovery and/or regeneration (see, for example, papers written by Kellmann et al.). Furthermore, I am still struggling to find the novelty of this study and for sure the authors did not properly highlight this point. This is really important as reviewers can better understand how a manuscript would enrich literature. What is likely contributing to my lack of understanding is the English adopted by the authors across the different sections. Indeed, the English language should be improved to ensure that an international audience can clearly understand the text. I suggest the authors contact a colleague who is proficient in English and familiar with the subject matter for reviewing the manuscript or contact a professional editing service. Participants are not well-described. Where does the sample come from? Still considering participants, I am totally aware of the difficulties most of the researchers encounter during recruiting processes, especially considering high-level athletes. However, I cannot justify the sample size here, because it is split into two groups, but especially because the nature of the paper, which should be considered as a case study, should have been explicated way earlier. On the contrary, this issue is simply treated as a limitation of the study. I was not able to clearly follow the procedure to assess sleep quality/parameters and the validity of the device used for this purpose should be fully referenced. Additionally, the authors reported the use of Cohen’s d for effect sizes but then they did not discuss their results considering them.
For all these reasons, I cannot recommend the manuscript for further revisions

Experimental design

No comment

Validity of the findings

No comment

·

Basic reporting

Dear Authors and Editors:
I appreciate your contribution to the preparation of this work, the topics covered in the text are interesting from the point of view of practical application. The protocol for accelerating post-exercise regeneration of footballers in professional leagues is a very important element of the work environment culture and also has great educational values, especially for young footballers. The application of appropriate rules may also reduce the risk of injuries, e.g. non-contact.
However, I have a few suggestions and recommendations that can improve the quality of your work and make it easier to interpret.


1. Basic reporting

The structure and language of the article are acceptable. I would suggest that the work be subjected to linguistic analysis again in terms of correctness and clarity, allowing the reader to easily understand its content. This will help to improve the quality of the language used in the article. I'll try to point out the mistakes below.

Line 23-25 - rewrite of clarity
Line 24 - change the verb form "is"
Line 25, 28, 38 - correct article usage "the"
Line 36 - correct article usage "a"
Line 38 - change preposition "of"
Line 50 - correct article usage "an"
Line 50 - change preposition "in"
Line 51 - change preposition "out"
Line 52, 57 - correct article usage "the"
And so, throughout the work, very similar errors appear. I found over 100 of them in the work. I do not list them all because the work requires redrafting in the introductory part and discussion.

2. Literature references, sufficient field background/context provided.
The construction of this part of the work is correct and indicates a specific research potential. Nevertheless, it needs to be redrafted about the literature in the discussion deviates from the accepted standard. Literature included in the discussion should be partly included in the introduction.

3. Professional article structure, figures, tables. Raw data shared.
No comments

4. Self-contained with relevant results to hypotheses.
No comments

Experimental design

1. Original primary research within Aims and Scope of the journal.
no comments

2. Research question well defined, relevant & meaningful. It is stated how research fills an identified knowledge gap.
no comments

3. Rigorous investigation performed to a high technical & ethical standard.
no comments

4. Methods described with sufficient detail & information to replicate.
Below are my comments:
Line 108 - Suggests adding a drawing to help other researchers map the test.
Line 130 - A reference to the literature confirming the correlation of the test with the estimated Vo2 max value at the end of the test should be added here. Unless the footballers were running around in masks with breath analysis. Indirect analysis (Vo2 max estimation) or direct analysis of the gases exhaled during the test.
Line 155 - There is also a lack of literature confirming the correctness and legitimacy of using such a division.
Line 161 - There is no description of the research procedure - when exactly blood was taken for analysis, how many people were at once, etc.
Line 169 - No reference to the literature - confirming the legitimacy of using such a classification.
Line 174 - Who and when collected such data (automatically through applications) or manually by the trainer?
Was sleep monitored only before the tests, or was the analysis daily and the result averaged? No methodological details about data collection

Line 201 - No information on how many players were simultaneously subjected to Yumeiho therapy, and how much time has passed since the end of the training session. The absence of such a framework makes it difficult to understand the methodology and apply it to other studies or club practices. There is no information on how many masseurs participated in the massage. There is also no information on what the players in group B were doing at the time when group A was subjected to additional measures.
The lactate value of people (competitors) from group B after a passive break equal to the time spent on massage in group A was not collected for the analysis. So we do not know if the additional treatment will speed up regeneration. This could be resolved in a discussion.
All this makes it very difficult or even impossible to repeat this experiment.

Validity of the findings

1. Impact and novelty not assessed. Meaningful replication encouraged where rationale & benefit to literature is clearly stated
The entire discussion section needs to be redrafted as it is closer to the introduction or literature review. There are not many places where authors compare their results with those of other authors of research papers. They do not analyze the reasons, which in no way leads to considerations on the correctness of the application (an interesting massage procedure), and so one cannot draw correct conclusions.

Details below:
Lines 290 - 305 The authors forgot to (refer to) interpret this information in relation to their results. In this case, it should be moved to the introduction, there is no element here about improving the aspects of post-exercise recovery.
Lines 306 - 311 VO2max value greatly impacts the speed and amount of post-workout regeneration. Especially max intensity efforts.
No confirmation or denial

Line 312 -315 - In this shape of the text we do not find a connection with the subject and purpose of the work. (no common ingredients)
Line 321-323 - This should be used to interpret progress in both groups A and B.
Line 324 -326 - This does not refer to the results shown in work - I do not see a connection with the assessment of proprioception - such an analysis was not performed.

Line 326-328 - This should be moved to the materials and methods section.

328 - 332 - Good paragraph - again no reference to the results obtained in the experiment.

Line 333 - 335 Is this more of a citation or a link to the results?

Line 338 - 340 - This may be a conclusion or suggestion for future research for authors or other researchers in the world—another part of the job.

Line 341 - 399 - This is not a discussion of the results of the experiment, but a preliminary analysis of the literature.

Because of the above, I suggest writing this chapter again from the beginning.

2. All underlying data have been provided; they are robust, statistically sound, & controlled.
The database and results are collected and presented correctly in the section and can be well interpreted in relation to the literature cited in the work.

3. Conclusions are well stated, linked to original research question & limited to supporting results.

Raw data files are provided and can be opened.

The Conclusions section needs reformulation - see comments above.

No information on funding and possible conflicts of interest

There is also no practical application in the daily work of trainers.

Additional comments

The article is not written in accordance with the art and canon of international scientific works - it requires changes. However, the database gives potential and opportunities to improve it. It has some quantitative limitations shown in the limitations section. I believe that after the corrections in the material and methods section and the reassessment of the discussion, it deserves another chance.
General note - the text in the work should be justified

I Wish You good luck in Your further research!
Michał Nowak, PhD
Jan Długosz University of Humanities and Natural Sciences in Częstochowa
Faculty of health sciences
[email protected]

Reviewer 4 ·

Basic reporting

The paper is interesting, as recovery methods after intense activities are currently being much discussed, however the work becomes a little confusing, and the authors do not clearly explain its design and neither the behavior of the control group , it would be interesting to rewrite the study design more clearly.

Experimental design

The authors made a sample calculation, because 8 we know that inferential statistics need a larger sample.

Describe how the groups were divided.

All tests were performed on the same day, do the authors not believe that this impairs the performance of the participants?

Describe the exact moments that Lactate was collected.

During the 20 weeks of the intervention, the athletes trained normally, explaining this issue and how they controlled the intensities.

The control group used other recovery methods, with only 8 athletes, do these different methods not confuse the study result?

Validity of the findings

The conclusions are interesting, however it is necessary to present the sample calculation to know the reliability of the results.

Additional comments

The paper is interesting but needs to make the corrections mentioned above, as they directly influence the results.

---

## Round 0.2 · accepted · Accept

The authors have addressed the reviewers' comments. This manuscript is ready for publication.

·

Basic reporting

Correction suggestions have been taken into account.

Experimental design

Correction suggestions have been taken into account.

Validity of the findings

Correction suggestions have been taken into account.

Additional comments

Most of the clues were added by the authors. Thanks to this, the work has acquired an acceptable shape and allows the information contained in the research to be made public.

Reviewer 4 ·

Basic reporting

The article topic is relevant.

Experimental design

Experimental design is right.

Validity of the findings

Validity